

# Selection and evaluation of reference genes for analysis of mouse *(Mus musculus)* sex-dimorphic brain development

Tanya T. Cheung, Mitchell K. Weston and Megan J. Wilson

Department of Anatomy, University of Otago, Dunedin, New Zealand

## ABSTRACT

The development of the brain is sex-dimorphic, and as a result so are many neurological disorders. One approach for studying sex-dimorphic brain development is to measure gene expression in biological samples using RT-qPCR. However, the accuracy and consistency of this technique relies on the reference gene(s) selected. We analyzed the expression of ten reference genes in male and female samples over three stages of brain development, using popular algorithms NormFinder, GeNorm and Bestkeeper. The top ranked reference genes at each time point were further used to quantify gene expression of three sex-dimorphic genes (*Wnt10b*, *Xist* and *CYP7B1*). When comparing gene expression between the sexes expression at specific time points the best reference gene combinations are: *Sdha/Pgk1* at E11.5, *RpL38/Sdha* E12.5, and *Actb/RpL37* at E15.5. When studying expression across time, the ideal reference gene(s) differs with sex. For XY samples a combination of *Actb/Sdha*. In contrast, when studying gene expression across developmental stage with XX samples, *Sdha/Gapdh* were the top reference genes. Our results identify the best combination of two reference genes when studying male and female brain development, and emphasize the importance of selecting the correct reference genes for comparisons between developmental stages.

## INTRODUCTION

Mammals that reproduce through sexual reproduction have intrinsic differences between male and female sexes. The most obvious morphological differences are those of reproductive organs. However, many other less apparent organs and tissues are also modulated in a sex-dimorphic manner. In particular, there exist numerous differences in the human brain between the male and female sexes. The rate of maturation of the female brain reaches its peak maturity at 10 years of age compared to the male brain, which only peaks 4 years later (*Giedd et al., 1997*). On average females have a higher percentage of gray matter although volume is smaller in comparison (*Goldstein et al., 2001*; *Gur et al., 1999*) neuronal cells in the hypothalamus may only be responsive to estrogen or testosterone (*Arnold, 2004*; *Lenroot et al., 2007*). Furthermore, numerous neurological diseases have sex-specific biases (*Fombonne, 2009*), including, schizophrenia (risk ratio 1 female:1.4

Corresponding author
Megan J. Wilson,
meganj.wilson@otago.ac.nz

male) (*Abel, Drake & Golds, 2010*; *Saha et al., 2005*) and Autism spectrum disorder (1 female:4 male) (*Mottron et al., 2015*). In order to understand how such differences arise at the level of gene expression, we set out to determine the best set of reference genes to study mouse sex-dimorphic brain development during key developmental stages.

The house mouse (*Mus musculus*) is an excellent choice for embryonic studies of mammalian development owing to its comparatively short gestation period and accelerated life span. In particular, mice have been widely used to study sex-dimorphic brain development (*Maekawa et al., 2014*; *Ngun et al., 2011*). To acquire more insight into the molecular drives of sex-dimorphic brain development, it is necessary to study the expression of genes in the developing brain. High-throughput sequencing technologies such as RNA sequencing (RNA-seq) provide a powerful technique to study changes in gene expression (*Hrdlickova, Toloue & Tian, 2017*). However, data gathered from these high-throughput technologies needs to be validated to ensure accurate interpretation through repeated biological replicates.

The most commonly used method for validating the expression of a gene identified by sequencing is Reverse Transcription quantitative Polymerase Chain Reaction (RT-qPCR). RT-qPCR allows for the detection and quantification of specific cDNA fragments generated from RNA samples. However, to obtain levels of expression comparable between samples, the target gene must be normalized to the expression of at least two internal controls (termed reference genes) that are stably expressed throughout all samples. Normalization is needed to compensate for different amounts of cDNA present in the sample along with differing PCR efficiencies of primer sets. Therefore, the selection of the reference gene is important as inappropriate reference genes can bias the data and thus lead to misinterpretation of results.

Ideally, the reference gene should be present at a consistent level across all compared samples, regardless of treatment or disease state of the sample. Furthermore, the chosen reference gene should be constitutively expressed across all cell types and tissues. However despite large-scale high-throughput technologies, no such gene has been found. Therefore the most common and validated approach is to find a reference gene that is the least variable in the specific context of the study.

The most regularly used reference genes are *Actb* and *Gapdh* (*Boda et al., 2009*). However, a number of studies have shown that these genes are expressed differentially in the brain. *Gapdh* has been shown to have sex dimorphic protein levels in adulthood (*Perrot-Sinal, Davis & McCarthy, 2001*) and to be up regulated in neuronal apoptosis (*Chen et al., 1999*; *Sawa et al., 1997*). *Actb* and *Gapdh* expression has also been shown to vary across tissue types, among cell types and also during stages of cell proliferation and development of the brain (*Sotelo-Silveira et al., 2008*; *Veazey & Golding, 2011*).

To fill in the gap in the literature, we set out to find a set of reference genes most suitable for studies on the embryonic brain from E11.5 to E15.5 in males and females. This work will prove highly valuable unbiased study to uncover suitable reference genes for studies of embryonic brain development and illustrate the importance of the accurate selection of reference genes for RT-qPCR analyses.

## METHODS

All animal work was performed under the University of Otago Animal Ethics Committee number: ET13/14. Inbred C57BL/6 mice were purchased from the Hercus Taieri Resource Unit (University of Otago, Dunedin, NZ).

### Sample collection

Whole embryonic brain tissue (overlying epidermis and cranial facial tissues were removed) and tail tips were collected from timed stages (E11.5 to 18.5) pregnant mothers. RNA was extracted from brain tissue using Purelink RNA mini kit (Ambion, USA) according to manufacturer's instructions. DNA was isolated from tail tips using 0.2 mg/mL (final concentration) Proteinase K (New England Biolabs, Ipswich, MA, USA) and then added to DirectPCR Lysis reagent 102-T (Viagen Biotechnologies, Los Angeles, CA, USA). Tail tips were incubated overnight at 55 °C and followed by heat inactivation of the proteinase K at 85 °C for 45 min. Samples were centrifuged at 14,000 g for 1 min to pellet cell debris and 2 µL of each sample was used for sexing PCR. The RNA for each time point and sex was a pooled sample, with a minimum of three separate biological samples collected for each condition (from separate litters). Total RNA was quantified with a Nanodrop (ThermoFisher, Waltham, MA, USA) and purity assessed using the 260/230 and 260/280 ratios. A ratio of ∼2 was taken as acceptable for pure RNA.

### Sexing of embryos by PCR

To sex the embryos, the *Sry* gene was amplified using primers listed in Table S1. Following PCR, products were run on a 2% agarose gel, a band appears at approximately 380 bp, indicating the presence of the Y-chromosome for male embryos.

### DNase treatment and reverse transcription

To remove genomic DNA present in the sample, 1 µg RNA was added to 1 µL DNase and incubated at 37 °C for 40 min. A phenol/chloroform extraction was carried out followed by ethanol precipitation to purify sample. Reverse transcriptase was performed using iScript (Bio-Rad, Hercules, CA, USA) according to manufacturer's instructions.

### RT-qPCR

Oligonucleotide primers (Table S1) were designed and ordered from Integrated DNA Technologies (IDT, Newark, NJ, USA). Each reaction contained 10 µL SYBR Select Master Mix (ThermoFisher), 1.25 µL 20 pmol forward and reverse primers, 6.75 µL water and 2 µL cDNA. All reactions also included a no reverse transcription control and each reaction was carried out in triplicate. RT-qPCR was carried out in the Stratagene Mx3000p (Agilent Technologies, San Diego, CA, USA) under the following conditions: denaturing at 50 °C for 2 min, annealing 96 °C for 2 min, followed by 40 cycles of amplification (96 °C 15 s, 60 °C 15 s and 72 °C 1 min). A final cycle for melt curve analysis was included for every qPCR plate (one cycle: 95 °C 1 min, 55 °C 30 s and 95 °C 30 s). The threshold was automatically set by the MXPro program (v. 4.10), and the Ct value for each sample was calculated from the average of three technical replicates.

## Data analysis

Data from RT-qPCR was then analyzed using three algorithms to calculate the most stable or best suited reference gene; NormFinder (*Andersen, Jensen & Orntoft, 2004*), GeNorm with SLqPCR R-based package (*Hellemans et al., 2007*) Bestkeeper (*Pfaffl et al., 2004*) and RefFinder (deltaCt method, (*Silver et al., 2006*; *Xie et al., 2012*) software). All data was analyzed according to the program instructions.

The geometric mean was calculated for each reference gene by assigning a number (1–10) for top ranked genes for each algorithm. The following equation was used to calculate the geometric mean for each data set: $\sqrt[3]{\text{NormFinder*GeNorm*BestKeeper*deltaCt}}$. The reference genes were then ranked again based on their lowest geometric mean.

Analysis of sex specific gene expression was calculated using the following equation: $\Delta Ct = 2^{-(Ct\text{gene}-Ct\text{reference})}$ with the average Ct value for the top two ranked reference genes. Statistical testing used was either the unpaired Student's $t$-test when comparing two time-points or a 2-way ANOVA when comparing multiple time-points across time (Tukey's multiple comparison test).

## RESULTS

### Selection of candidate reference genes and mRNA transcript levels

To ensure accurate analysis of gene expression, ten candidate reference genes (Table 1) were investigated to determine how stable each gene is for sex and time-point of development. The candidate reference genes selected included those genes commonly used in RT-qPCR experiments (such as *Gapdh* and *Actb*) and genes were stably expressed in multiple mouse adult tissues, but had not been tested for suitability as reference genes with mouse embryo tissues (*Kouadjo et al., 2007*). RT-qPCR was carried out for brain tissue samples across three time points: E11.5, E12.5 and E15.5 using both male and female samples. These time points were chosen for analysis as the sex determination in mouse occurs between E11.0 and E12.0. During this 24 h period, considerable changes also occur with respect to neuronal development, in particular, the formation of the primary brain vesicles (*Stiles & Jernigan, 2010*). By E12.5, cellular proliferation is increased resulting in the expansion of neuronal precursors and formation of cortical layers (*Finlay & Darlington, 1995*). Neuronal differentiation, axonal branching and synaptogenesis is taking place around day 15.5 (summarized in Fig. 1) (*Sur & Rubenstein, 2005*) The raw Ct values for each reference gene across time and each sex are plotted in Fig. 2.

### Determination of best-suited reference genes for male and female developing brain

Four algorithms: NormFinder, GeNorm, BestKeeper and comparative deltaCt were employed to determine the best-suited reference genes for either specific developmental stage or between sex over time. Female and male samples were combined to identify the most stable reference gene across time. At each developmental stage (E11.5, E12.5 and E15.5) female and male samples were compared against each reference gene to determine the best reference gene between both sexes.

**Table 1  Function, symbol and name of selected reference genes.**

| Gene symbol | Gene description | Function |
|---|---|---|
| Gapdh | Glyceraldehyde 3-phosphate dehydrogenase | Catalyzes sixth step of glycolysis |
| Actb | Beta-actin | Formation of microfilaments in eukaryotic cells |
| Hprt1 | Hypoxanthine guanine phosphoribosyl transferase | Transferase that plays a role in the generation of purine nucleotide through the purine salvage pathway |
| Pgk1 | Phosphoglycerate kinase 1 | Part of the glycolysis pathway which catalyses the conversion of 1,3-diphogycerate to 3-phosphoglycerate |
| Sdha | Succinate dehydrogenase complex, subunit A, flavoprotein (Fp) | Citric acid cycle and the respiratory chain |
| Ppia | Peptidylprolyl isomerase A | Catalyzes cis-trans isomerization of proline imidic peptide bonds. Role in protein folding |
| RpL38 | Ribosomal protein L38 | Protein synthesis |
| RpL37 | Ribosomal protein L37 | Protein synthesis |
| Eif3f | Eukaryotic translation initiation factor 3 subunit F | Translation elongation |
| Eef2 | Eukaryotic translation elongation factor 2 | Translation elongation |

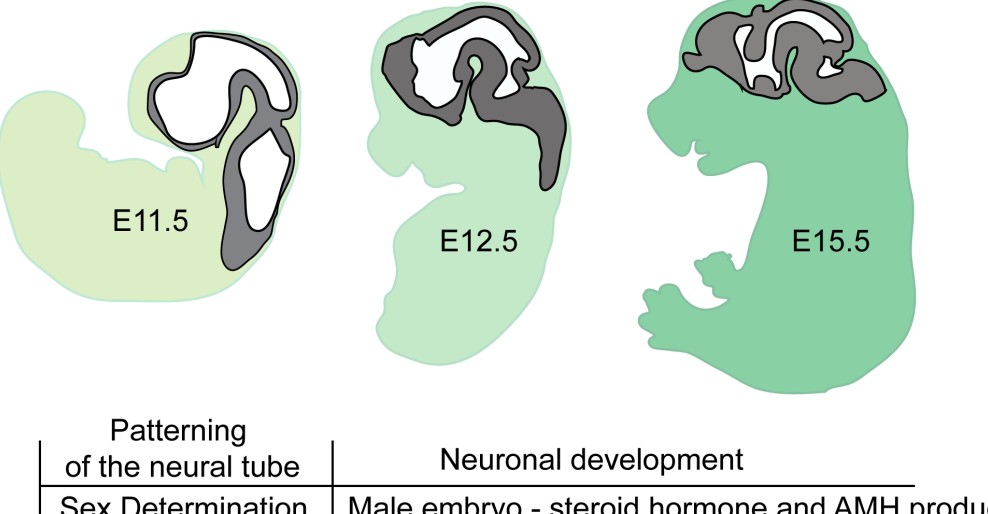

**Figure 1  Brain developmental stages used in this study.** In mice gonadal sex determination occurs at ∼E11.5. At E11.5 the neural tube has formed the primary brain vesicles: Prosencephalon, Mesencephalon and Rhombencephalon. Between stages of E12.5 and E15.5, there is increased expansion of neuronal precursors and cell migration forming cortical layers. Simultaneously, neurons differentiate to allow for axonal branching and synapse formation.

The raw Ct values for each gene comparing sex-expression at each time point are shown as box and whisker plots (Fig. 2) with minimum and maximum values indicated at each time point. Ct values for male samples ranged between Ct values of 15–25.5 and female samples between 15–28 (Table S3). Across all time points, the reference gene with least amount of variability for male samples is *Actb* with a mean Ct value of 16.36 (±1.178 SD) while the highest is *Eif3f* with a mean Ct value of 30.06 (±5.08 SD). In females, *Actb* mean

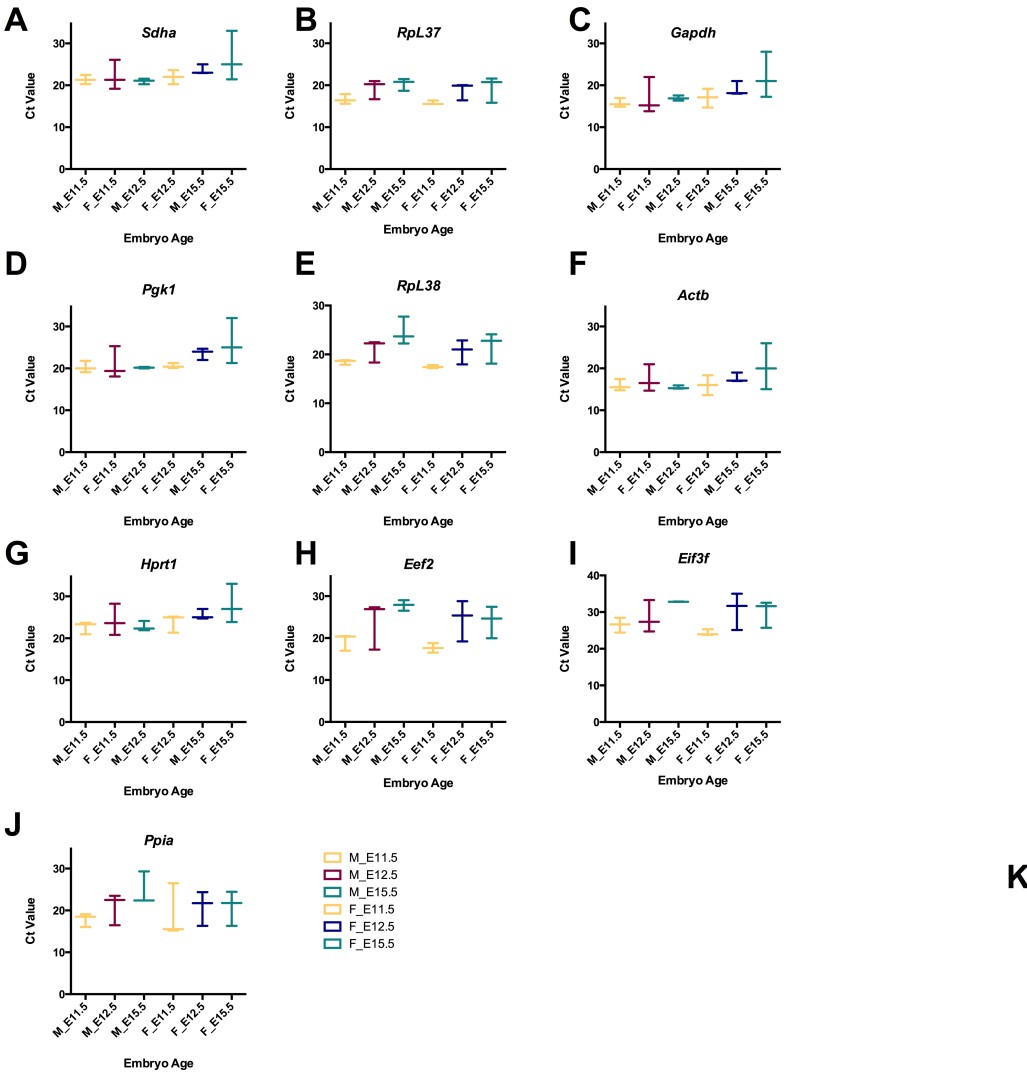

**Figure 2   Raw Ct values for selected reference genes from RT-qPCR using cDNA from male and female brain tissue.** (A) Raw Ct values for each reference gene are shown for female and male samples by developmental stage. Data is plotted as floating bar plot, with min and maximum values indicated and a line at the mean for each stage (E11.5, E12.5 and E15.5).

Ct value is 17.9 (±2.22 SD) and *Eef2* has a mean Ct of 22.53, while this Ct value was not the highest, the SD had the highest amount of variability of ±4.54 SD.

NormFinder Excel based add-on is an algorithm created by *Andersen, Jensen & Orntoft (2004)* that ranks a set of candidate reference genes using stability values according to the variation in expression across samples and between groups. A low stability value represents a reference gene with the most stable expression in a given sample set. NormFinder analysis identified, that across all developmental stages of development (Fig. 3A, Table S4), the reference gene with the lowest stability value was *RpL37* (1.211), the best combination of genes were *Hprt1* and *RpL38* (0.559) in the male embryonic brain. In female samples, *Hprt1* (0.929) was the most stable gene across all time points, the best two gene combination

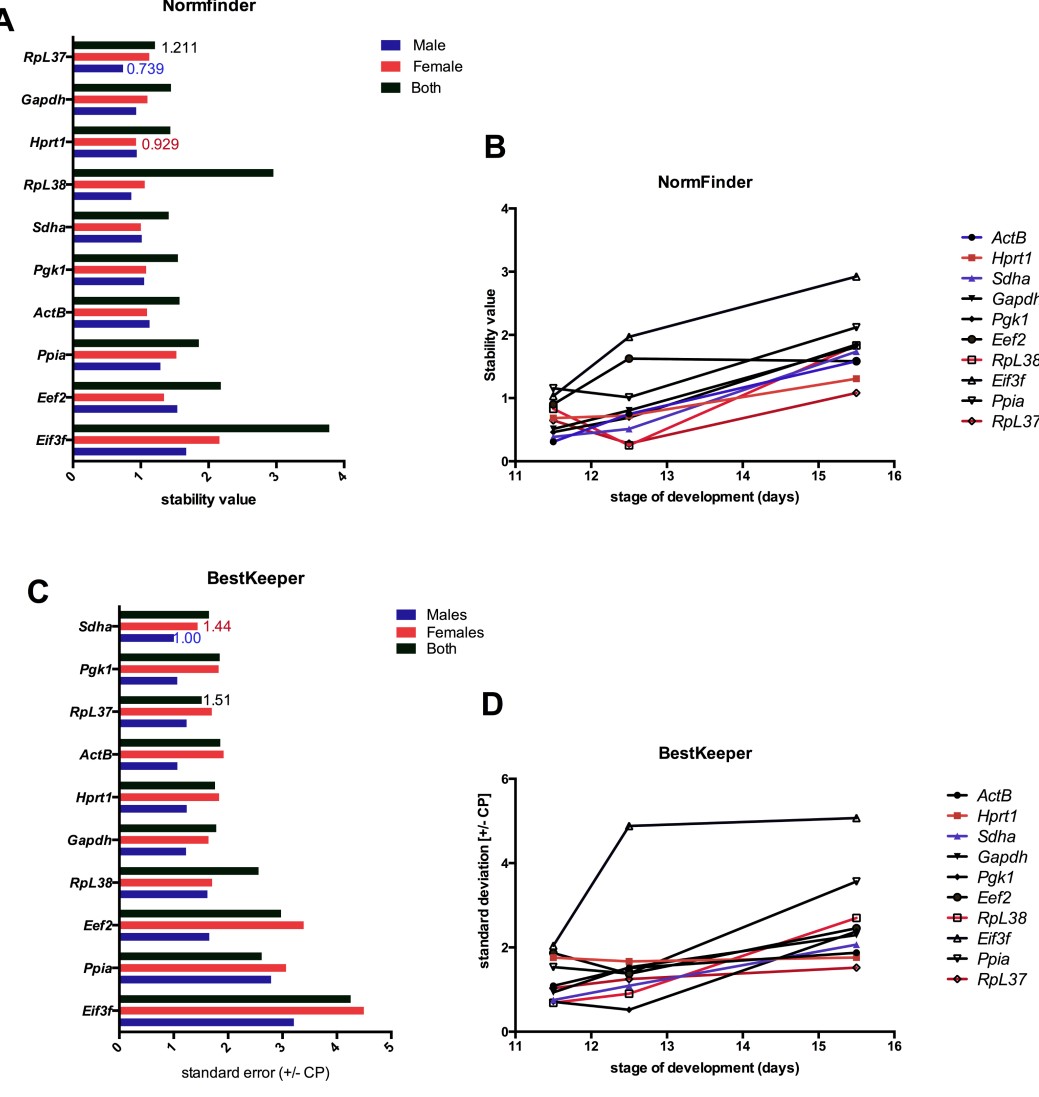

**Figure 3** **Comparison of gene stability values for NormFinder and BestKeeper.** (A) NormFinder stability values for male and female reference genes across all developmental stages. (B) The stability value of combined sexes calculated by NormFinder across all each developmental time point of development. (C) The best suited reference genes for male and female samples as calculated by BestKeeper shown as sdt error (±CP). (D) BestKeeper values shown as std error (±CP) across time in samples of both sexes. The lowest stability value for each sex is shown (Female, red; Male, blue; Both sexes, black).

was *Sdha* and *RpL38* (stability value of 0.640) (Fig. 3A). NormFinder identified the lowest stability value at E11.5 to be *Actb* (0.309) (Fig. 3B), however *Actb* and *Sdha* (0.250) genes gave the best combined stability value (0.250) for RT-qPCR analysis. Following sex determination of the embryo, *RpL38* (0.253) was the most stable reference gene at E12.5, with a combination of *RpL38* and *RpL37* (0.189) for use of two reference genes (Fig. 3B). *RpL37* (0.145) is the most stable reference gene at E15.5 however for analyses with two reference genes then *Pgk1* and *Eef2* (0.430) were recommended (Table S3).

BestKeeper software determines the ideal reference genes out of a number of candidates based on the standard error (±CP) of Ct values and combines them into an index using repeated pair-wise correlation analysis (*Pfaffl et al., 2004*). Across all developmental stages, the BestKeeper identified *Pgk1* (SE: ±1.064) and *Sdha* (SE: ±1.004) as the best reference gene in the male sample, while *Gapdh* (SE: ±1.639) and *Sdha* (SE: ±1.441) to be ideal for female samples (Fig. 3C). However at specific developmental stages, the ideal reference gene differs. At E11.5 *RpL38* (SE: ±0.68) is ranked the top reference gene, while *Pgk1* (SE: ±0.522) and *Hprt1* (SE: ±1.762) are the most stable reference genes for E12.5 and E15.5 respectively (Fig. 3D, Table S5).

The SLqPCR R-based package uses the GeNorm algorithm (*Hellemans et al., 2007*) which determines the most stable reference gene from a set of samples by calculating the geometric mean of each reference gene in a stepwise calculation. A low M value indicates stable expression of the gene: for homogenous samples a value below 0.5 indicates an unstable reference gene in the samples analyzed, whereas for heterogeneous tissues the mean M value below 1 is acceptable (*Hellemans et al., 2007*). The best stability value for the female samples across all time points was *Sdha* and *Gapdh* with a stability M value of 0.591 while the top three most stable reference genes in the male samples were *Pgk1,* and *ActB* with a mean M stability value of 0.281 (Fig. 4A, Table S6).

At specific developmental time points, *Sdha* and *Pgk1* (M of 0.2470) were calculated as the most stable reference genes for analysis of male and female samples at E11.5. Following sex determination in both male and female samples, *Actb* and *Gapdh* (M of 0.346) gene are the most stable (Fig. 4B). During increased neuronal proliferation at E15.5, *Actb* and *Pgk1* (M of 0.719) gave the lowest M values, indicating these are reliable reference genes according to GeNorm analysis given at this stage the developing brain is heterogeneous (Table S6).

A forth method, termed the comparative deltaCt method (*Silver et al., 2006*) was also used to compare reference genes. This uses a method similar to that of GeNorm, determining the variation of gene expression between paired putative reference genes (the deltaCt) within each sample. Variable deltaCt values for each pair between multiple samples, resulting in a high standard deviation when the average deltaCt is calculated, means either gene or both are not stably expressed. Pairs at E11.5 were *Pgk1* and *Sdha* (1.98/1.87), *Actb* and *Pgk1* (1.98/1.82). At E15.5 there was much more variation between reference genes, best pair were *Actb* and *Sdha* (4.7) (Fig. 4D; Table S7) When considering brain development overtime, for the male samples *Pgk1* and *Sdha* (2.1) were the best pair, whereas for female samples, *Hprt1* and *Sdha* (2.7) had the lowest stand deviation of deltaCt values (Fig. 4C).

## Geometric mean of each reference gene

Each reference gene was ranked from most stable to least stable, as determined by the four stability calculations described above, and assigned a number from 1 (most stable) to 10 (least stable) (Table S8). The geometric mean was taken for each set of rankings to calculate an overall 'best' reference gene for each condition (Tables 2 and 3). The recommended reference genes across all stages in both male and female samples are *Sdha* and *RpL37*

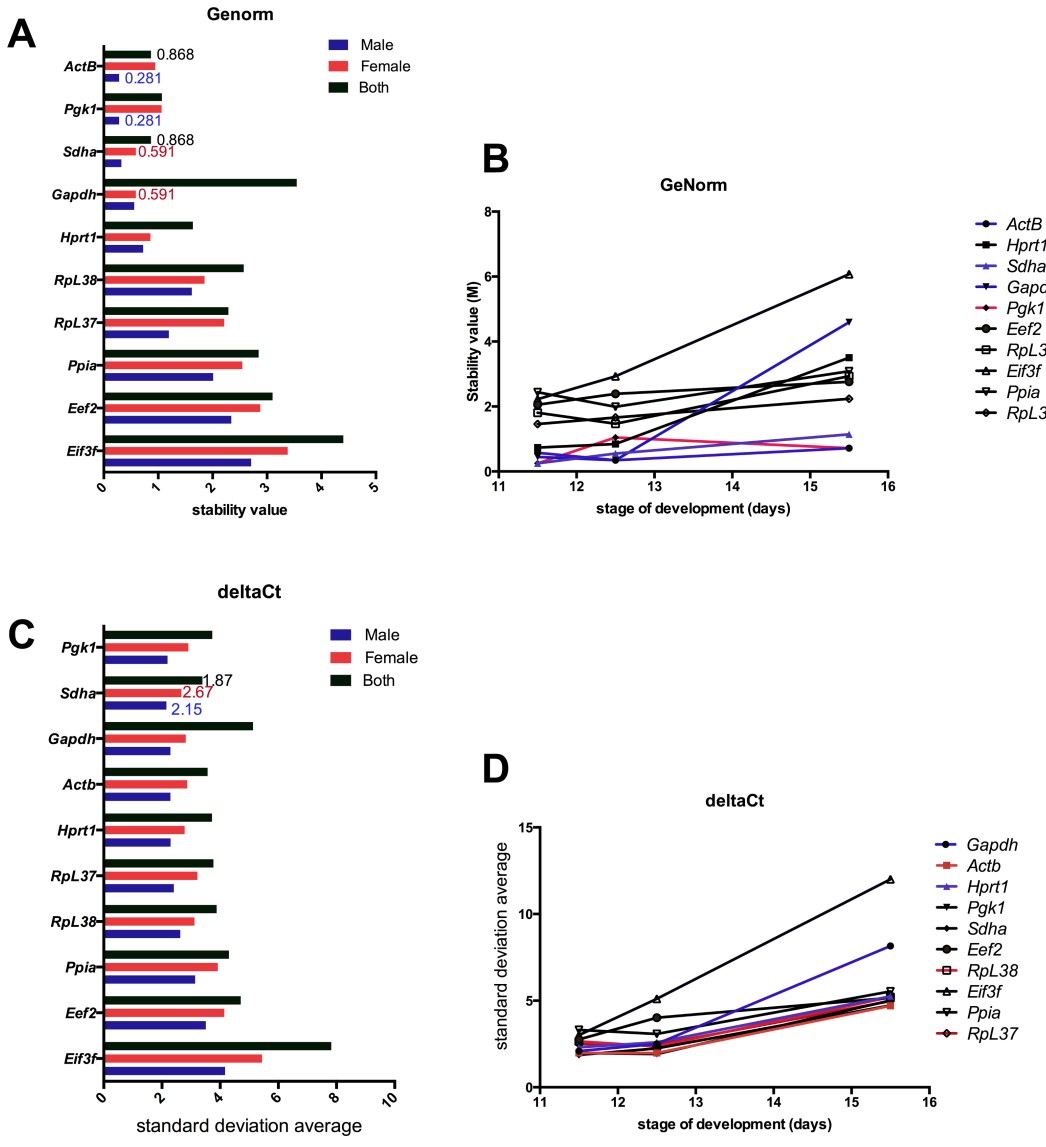

**Figure 4** **Comparison of stability values for GeNorm and deltaCt methods.** (A) Stability (M) value calculated by GeNorm across time in both male and female samples. (B) GeNorm mean stability (M) values in male and female samples across all developmental stages. (C) Average standard deviation using the deltaCt method for male, female and combined samples. (D) Average standard deviation using the deltaCt method for each time point tested. The lowest stability value for each sex is shown (Female, red; Male, blue; Both sexes, black).

(GeoMean: 1.18 and 2.94 respectively). Across all stages of embryonic development in males, the recommended reference genes are *Actb* and *Sdha* (GeoMean: 2.73 and 1.96) while in the female brain, the most stable set of reference genes are *Sdha* and *Gapdh* (GeoMean 1.2 and 2.34).

For RT-qPCR analyses at specific developmental time points regardless of sex (Table 3), *Sdha* and *Pgk1* (GeoMean: 1.56 and 1.86) are best suited for analysis at E11.5. Following determination of sex in the embryo at E12.5, *RpL38* and *Sdha* with GeoMean values of 1.68

**Table 2 Geometric Mean of ranking values.**

|  | Male | | Female | | All stages, male and female | |
|---|---|---|---|---|---|---|
| 1 | Sdha | 1.96 | Sdha | 1.20 | Sdha | 1.18 |
| 2 | Actb | 2.73 | Gapdh | 2.34 | RpL37 | 2.94 |
| 3 | Pgk1 | 2.91 | Hprt1 | 2.45 | Actb | 3.03 |
| 4 | Gapdh | 3.78 | RpL37 | 4.58 | Pgk1 | 3.10 |
| 5 | RpL37 | 4.16 | Actb | 4.60 | Hprt1 | 3.83 |
| 6 | Hprt1 | 4.64 | Pgk1 | 5.23 | RpL38 | 6 |
| 7 | Actb | 4.82 | RpL38 | 5.63 | Gapdh | 6.34 |
| 8 | Ppia | 7.95 | Ppia | 8.23 | Ppia | 7.23 |
| 9 | Eef2 | 8.27 | Eef2 | 8.73 | Eef2 | 8.30 |
| 10 | Eif3f | 10 | Eif3f | 10 | Eif3f | 10 |

**Table 3 Geometric Mean of ranking values by developmental stage.**

|  | E11.5 | | E12.5 | | E15.5 | |
|---|---|---|---|---|---|---|
| 1 | Sdha | 1.56 | RpL38 | 1.68 | RpL37 | 1.86 |
| 2 | Pgk1 | 1.86 | Sdha | 1.73 | Actb | 1.86 |
| 3 | Actb | 2.91 | Pgk1 | 3.31 | Sdha | 3.31 |
| 4 | Gapdh | 3.72 | Actb | 3.35 | Pgk1 | 3.72 |
| 5 | RpL38 | 4.3 | RpL37 | 3.89 | Hprt1 | 3.86 |
| 6 | RpL37 | 5.48 | Gapdh | 4.09 | Eef2 | 4.78 |
| 7 | Hprt1 | 5.89 | Hprt1 | 5.95 | RpL38 | 6.70 |
| 8 | Eef2 | 8.23 | Ppia | 7.44 | Gapdh | 7.02 |
| 9 | Ppia | 9.14 | Eef2 | 7.77 | Ppia | 8.20 |
| 10 | Eif3f | 9.24 | Eif3f | 10 | Eif3f | 10 |

and 1.73 are recommended. Finally, *RpL37* and *Actb* (GeoMean 1.86 and 1.86) are the best suited for analysis of E15.5 brain tissue.

In comparison to a careful sex or stage approach above, we also pooled all the data together (both sexes and all stages). Overall, this approached produced stability values that were high (indicating a great variation between samples) for GeNorm and BestKeeper packages (Tables S5 and S6). The geometric mean of ranked references genes for a pooled sample approached suggested that *Sdha* and *RpL37* (Tables 2 and 3), despite *RpL37* often being ranked a poor choice if we were just consider the stabilities values for E11.5 and E12.5 individually (Table S8 and S3).

## Confirmation of sex-specific gene expression

Previous micro-array data showed sex-specific expression of a number of genes within the embryonic brain (forebrain) at E10.5 (*Dewing et al., 2003*). *Wingless-type MMTV integration site family, member 10B* (*Wnt10b*) was shown to be up-regulated in male embryos, with 1.7 fold increase in expression at E10.5 with respect to the female brain. Cytochrome P450,7b1 (*CYP7B1*) was expressed 2.1 fold higher in the male brain, whereas expression of X-inactive specific-transcript (*Xist*) was confirmed significantly higher in

the female embryo with a fold change of 18.5 compared to male samples. However, these differences in gene expression were not studied over further time points, when neurogenesis has commenced. To extend the findings from *Dewing et al. (2003)*, regarding sex-dimorphic expression of these genes, the top two ranked reference genes at each time point from this study are used to compare against a selection of three genes in male and female samples across embryonic brain development.

To normalize the expression of our three genes of interest, *Sdha* and *Pgk1* oligonucleotide primers were used at E11.5, *RpL38* and *Sdha* were used for analysis for E12.5. For analysis of E15.5, the reference genes *Actb* and *RpL37* were used. Sex-dimorphic expression of *Xist* was detected at all three time points with increased expression in female brain compared to the males (Fig. 5A). *CYP7B1* and *Wnt10b* also showed significant sex-dimorphic expression at E11.5 ($P < 0.05$), with higher expression observed in the male brain at this stage (Figs. 5C and 5E; 3.5-fold and 5-fold respectively). Additionally, following sex determination, *Wnt10b* and *CYP7B1* expression was not significantly different between the sexes using these stage-specific reference genes (Figs.5C and 5E). In contrast, when pooling all data from sex and age together the top two ranked genes were *RpL37* and *Sdha*. When using these two genes to normalize expression data there is a large variation in the expression of the three genes, especially at E12.5, between biological replicates (Figs. 5B, 5D and 5F). No significant differences were observed in gene expression between male and female samples, when using *Pgk1* and *Sdha* as reference genes with the exception of *Xist* expression at E12.5 (Figs. 5B, 5D and 4F).

When studying gene expression changes over time, two slightly different sets of reference genes were ranked highest, depending upon the genotype of the sample (Table 2). We normalized gene expression at each time point for female samples (with *Sdha* and *Gapdh*) and for male samples (with *Pgk1* and *Sdha*) to study how expression of these genes changes with respect to developmental age (Fig. 6). In comparison, when *RpL37* (one of the poorer stable genes for female and male time points) and *Sdha* where used as reference genes for normalization of the all data points (top two genes when reference gene data was pooled Table 2), this introduced so much variation between samples, any analysis would be inconclusive (Fig. S1).

*CYP7B1* mRNA expression declines between E11.5 and E12.5 in both sexes but its expression significantly increases again in the female only by E15.5 (Fig. 6A, $P < 0.05$). *Xist* transcript expression in the female brain also increased between E11.5 and E15.5 (following sex-determination window (Fig. 1)) (Fig. 6B; $P < 0.001$). Gene expression of the *Wnt10b* gene in the developing brain was low at all time-points and did not change significantly overtime in the female staged embryo samples (Fig. 6C). However, the higher expression observed in the male embryonic samples decreased also between E11.5 and E12.5, to similar expression levels found in XX samples (Fig. 6C).

## DISCUSSION

While studies using RTq-PCR have proven to be a powerful tool to study changes in gene expression, there must be a careful consideration of the type of reference genes to be used.

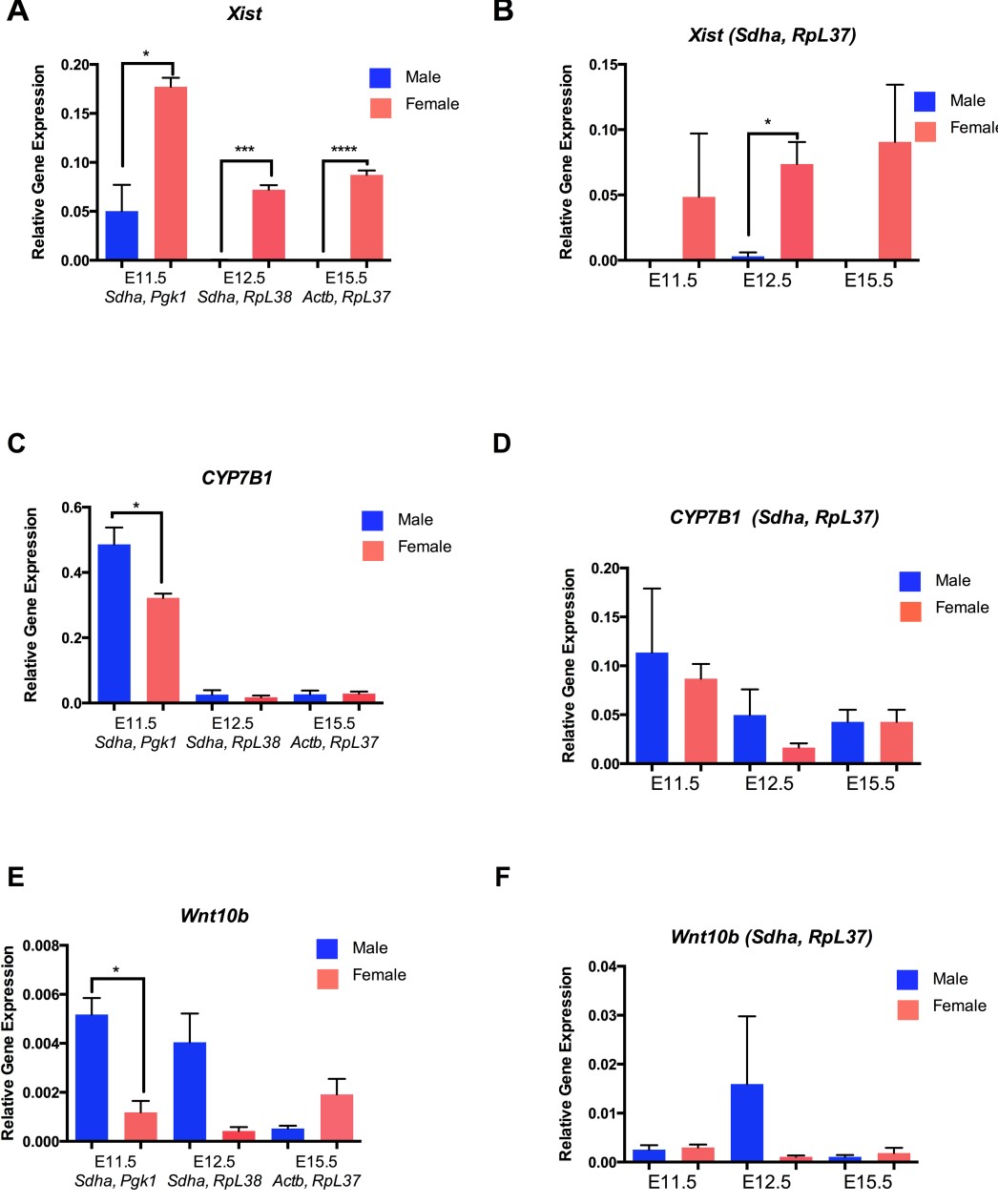

**Figure 5  Sex specific gene expression using top ranked reference genes.** (A) *Xist* expression compared to the mean of the top two reference genes at each developmental time point. (B) *Xist* expression compared to *Pgk1* and *Sdha* reference genes at all stages of development. (C) *CYP7B1* expression compared to the mean of the top two reference genes at each developmental time stage. (D) *CYP7B1* expression compared to *Pgk1* and *Sdha* reference genes at all stages of development. (E) *Wnt10b* expression compared to the mean of top two reference genes at each developmental time point. (F) *Wnt10b* expression compared to *Pgk1* and *Sdha* across all stages of development. Bar graphs are shown as the mean of three replicates with error bars as the mean standard error of the mean. * = $P < 0.05$ ** = $P < 0.001$ *** = $P < 0.0001$ (Student's unpaired $T$-test).

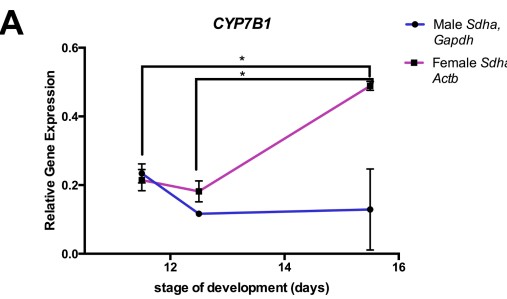

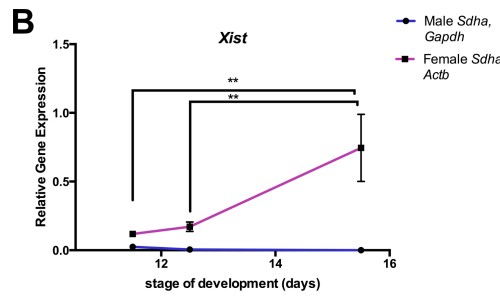

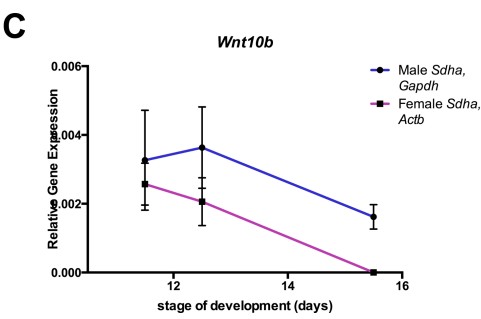

**Figure 6** **Expression of *CYP7B1*, *Xist* and *Wnt10b* across three stages of brain development.** (A) *CYP7B1* mRNA expression in male (blue) and female (pink) brain tissue across developmental stages E11.5, E12.5 and E15.5 normalised to the top ranked reference genes for each sex over time. (B) *Xist* mRNA expression in male (blue) and female (pink) brain tissue across developmental time stages normalized to the top ranked reference genes for each sex at E11.5, E12.5 and E15.5. (C) *Wnt10b* mRNA expression of male (blue) and female (pink) brain tissue at developmental time stages E11.5, E12.5 and E15.5. Data is show as mean and standard error of the mean. * = $P < 0.05$ ** = $P < 0.001$ Two-way ANOVA (Tukey's multiple comparison test).

As shown here, the reference genes used at particular developmental stages introduce much more variation into the data analysis and is problematic for drawing conclusions from RT-qPCR data. Therefore, it is essential to determine use the best reference gene pair for RT-qPCR analyses. For the comparison of sex-dimorphic gene expression at specific time points, *Sdha* and *Pgk1* are recommended at E11.5. At E12.5, *Actb* and *Sdha* are most appropriate followed by *Gapdh* and *Pgk1* are E15.5.

We determined the expression across time and between male and female sexes for three genes, previously identified as being expressed sex-dimorphically prior to sex-determination using the highest ranked references genes for each sex (*Dewing et al., 2003*). Here we identified the ideal reference genes when studying male gene expression across time, to be a combination of *Actb* and *Sdha*. In contrast, when studying gene expression across developmental stage with female samples, *Sdha* and *Gapdh* would be most appropriate. In comparison, if we used the *Pgk1* gene (an X-linked gene) as the second reference for female samples, this introduced a lot of variation between biological replicates ($n = 3$, pooled samples from different litters) at E11.5 and E12.5. This suggests that expression of this commonly used reference gene was not stable in female embryonic brain tissues (Fig. S1) *Pgk1* and *Hprt* are X-linked genes, subject to X inactivation during cell differentiation

(*Chaumeil et al., 2006*; *Heard, Clerc & Avner, 1997*). This indicates that X-linked genes should be avoided when analyzing gene expression in XX tissues samples at early stages of brain development, particularly as our data for *Xist* (Fig. 6) indicates changes to the expression of X-linked genes during brain development.

CYP7B1 converts the dihydrotesosterone (DHT) metabolite $5\alpha$-androstane-$3\beta$, $17\beta$-diol ($3\beta$-diol) to an inactive form and has been suggested to have a developmental role maintaining normal levels of estrogens and androgens in the mammalian brain (*Rose et al., 1997*). *CYP7B1* mRNA expression has been previously detected in the hippocampus, testis and ovary (*Wu et al., 1999*). Knockout *CYP7B1* mice have an enlarged brain (late fetal stages were examined) due to reduced apoptosis but overall brain weight normalizes after puberty for both sexes (*Sugiyama et al., 2009*). In female mice *CYP7B1* knockout results in early onset puberty and ovarian failure (*Omoto et al., 2005*). A recent study found that male mutant have a reproductive behavioral defect, possibility due to altered olfactory cue sensing (*Oyola et al., 2015*). We found that expression of *CYP7B1* is sex-dimorphic at E11.5, prior to any steroid hormone production by the developing gonads. By E15.5 the testis is producing significant levels of testosterone and DHT, expression levels of *CYP7B1* remained low in the male but had increased in the female at E15.5 (Fig. 4C). Consistent with these results, previously it was found that the human *CYP7B1* promoter is suppressed by DHT in human cell lines (*Tang et al., 2006*). In contrast, *CYP7B1* expression increased with overexpression of estrogen receptors. While in females there is no prenatal estradiol production, $3\beta$ Adiol, the target of CYP7B1 activity, is produced by the immature ovary (*Sugiyama et al., 2010*). $3\beta$ Adiol binds to estrogen receptor $\beta$ (ER$\beta$), expression of this receptor begins at E12.5, peaks at E18.5 and is active in the absence of estrogen, possibility due to the presence of an alternative ligand such as $3\beta$ Adiol (*Sugiyama et al., 2010*). Together, this indicates that changes to the embryonic brain gene expression of *CYP7B1* maybe regulated by androgen receptors and levels of steroid hormone metabolites during mouse development. This may produce the changes we observed in *CYP7B1* expression across time in the female and males developing brain RNA samples (Figs. 4C and 5A).

The Wnt ligand, *Wnt10b*, was expressed at higher levels in male samples compared to female E10.5 head samples in a previous microarray study (*Dewing et al., 2003*). While *Wnt10b* knockout mice appear phenotypically normal, they exhibit a number of aging-related phenotype such as bone loss (*Stevens et al., 2010*). Little has been studied regarding the role of *Wnt10b* in early brain patterning, however research with the zebrafish model indicated that *Wnt10b* functions redundantly with *Wnt1* in specification of the midbrain-hindbrain boundary (*Lekven et al., 2003*). Therefore it may play an early role in patterning of the midbrain-hindbrain boundary in mice, but any functional consequence for higher expression within the male developing brain at E10.5 (*Dewing et al., 2003*) and E11.5 (Fig. 4E) requires further investigation.

*Xist* is an essential long non-coding RNA that has a role in gene dosage compensation in XX embryos, by X chromosome in activation (XCI). During inactivation, the X chromosome that will be inactivated up-regulates *Xist* expression, whereas *Xist* gene expression from the active X chromosome is repressed (*Galupa & Heard, 2015*). We observed high levels of *Xist* expression in the female brain, compared to low or undetectable

levels of *Xist* expression in the males between E11.5-15.5 (Fig. 4A). Regulation of *Xist* gene expression is complex, it is repressed by promoter bound CCCTC-binding factor (CTCF) and negatively regulated by an anti-sense RNA, *Tsix*. A second ncRNA transcript, *Jpx*, expressed from both X chromosomes, binds to CTCF and removes it from the *Xist* promoter region promoting expression of *Xist* (Sun et al., 2013). *Xist* gene expression (nor that of *Jpx* and *Tsix*) has not been examined in any detail overtime nor in later developing tissues. The function of *Xist* RNA in XCI is likely to be dosage sensitive (to *Xist* RNA levels) and tightly regulated. Mis-regulation (up- or down) will allow either more genes to escape X-inactivation or result in a reduction of expression of X-linked genes that are critical for brain development. Overexpression of *Xist* has been linked to female psychiatric conditions in humans (Ji et al., 2015).

This study has shown that the optimal reference gene(s) varies with the sex and stage of development. Therefore, when looking at gene expression either between sexes or across time for a particular sex, careful consideration should be given to which reference gene(s) are the most stable between the given samples and ideally use a combination of at least two to provide the most robust results for data analysis.

## ACKNOWLEDGEMENTS

We would like to thank Simon Blanchoud for the very helpful comments on the manuscript drafts and Lisa Zondag for proofreading of the manuscript.

### Funding

The authors received funding from the Otago Medical Research Foundation. The funders had no role in study design, data collection and analysis, decision to publish, or preparation of the manuscript.

### Grant Disclosures

The following grant information was disclosed by the authors:
Otago Medical Research Foundation.

### Competing Interests

The authors declare there are no competing interests.

### Author Contributions

- Tanya T. Cheung conceived and designed the experiments, performed the experiments, analyzed the data, contributed reagents/materials/analysis tools, wrote the paper, prepared figures and/or tables, reviewed drafts of the paper.
- Mitchell K. Weston performed the experiments, contributed reagents/materials/analysis tools, reviewed drafts of the paper.
- Megan J. Wilson conceived and designed the experiments, analyzed the data, contributed reagents/materials/analysis tools, wrote the paper, prepared figures and/or tables, reviewed drafts of the paper.

## Animal Ethics

The following information was supplied relating to ethical approvals (i.e., approving body and any reference numbers):

University of Otago Animal Ethics Committee ET13/14.

## Data Availability

Data are supplied as Tables S1–S8.

## Supplemental Information

Supplemental information for this article can be found online at http://dx.doi.org/10.7717/peerj.2909#supplemental-information.

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
