# Peer review of "Selection and evaluation of reference genes for analysis of mouse (Mus musculus) sex-dimorphic brain development"

_PeerJ, doi:10.7717/peerj.2909_

## Round 0.1 · original submission · Major Revisions

After a delay for which we apologize, we have successfully obtained an informed external review, and I have provided review-level comments of my own in an attached document. Please respond to all points of both reviews if you choose to revise and resubmit.

I agree that your article is well-written but it would have more impact and significance; and differentiate itself better from similar work, with the suggested changes and discussion.

Thank you again for submitting your work to PeerJ.

Reviewer 1 ·

Basic reporting

This is a well-written manuscript describing the methods used for evaluation of the reference genes. The conclusions are important and should be of benefit to those using Q-RT-PCR in this field.See general comments for the author.

Experimental design

The manuscript has clear objective and the experimental design has been carried out with the ethical standards. See general comments for the author.

Validity of the findings

Data are robust, statistically sound, and controlled. See general comments for the author.

Additional comments

In this manuscript Tanya Cheung & Megan Wilson evaluated a panel of genes to find the suitable reference genes for normalization in quantitative RT-PCR analyses of Mus musculus sex-dimorphic brain development. The rationale behind their work is based in the observation that genes used as reference controls can vary based on cell type and conditions, thus making comparisons within and across publications very difficult. Therefore, the aim of the current manuscript is to demonstrate the best set of the suitable reference genes for normalization in quantitative RT-PCR analyses.
The demonstration of variability associated with commonly used reference genes, including Gapdh & Actb, is not novel – in other fields of biology this has led to similar efforts to identify better reference genes. With RNAseq becoming more common, large-scale datasets will inevitably provide a more stable platform for determining relative, and possibly even absolute, gene expression profiles. Nevertheless, validity qRT-PCR studies will still be required, and this technical report would represent the first survey to identify more stable reference genes in a Mus musculus sex-dimorphic brain development.
The main issue related to this article is the number of reference genes used. The authors tested only five reference genes, and over the five, three of them (Gapdh, Actb, Hprt) are commonly used and are problematic due to their instability in different mice models and tissues as showed in many published articles. Therefore, since the main goal of this report is to find the best genes for the use as a qRT-PCR reference, it is perplexing why only five reference genes are tested in this model.
The current availability of different algorithms (i.e. normfinder, geNorm, best keeper) to rank the tested reference genes against each other and to show the highest stable and lowest stable genes, gives the authors the choice to test more than 5 genes; 5 genes being not enough to make strong conclusions. Therefore, to strengthen their manuscript, it is important and highly critical for the authors to address this concern as part of a substantially revised version by adding at least 5 additional reference genes. Kouadjo et al 2007 (PMCID: PMC1888706) reported a wide range of housekeeping genes in mouse tissues. In that article the authors can find many reference genes that can be tested in their model to improve their manuscript.

Additional comments:
• The authors have to test the primer specificity and efficiency for each potential gene. They did not show the primer efficiency in the tested model. Efficiency % and R2 need to be shown.
• Three algorithms used are enough to rank the potential genes, but it would be great if the authors could calculate and show the overall comprehensive ranking to demonstrate the best gene.
• Figure 2: The authors tried to show Raw Ct values for selected reference genes from RT-qPCR using cDNA from male and female brain tissue. But, the authors did not test if there is a statistical difference between the different tested groups for each gene. It is highly recommended to do statistical analysis for raw ct values.
• Figure 3: the authors have to simplify their data presentation by showing the data in order such as start with high stable gene and end with least stable gene for each algorithm.

---

## Round 0.2 · accepted · Accept

Thank you for adding the requested improvements to the original submission. Looking forward to seeing this helpful work appear in our journal.

Reviewer 1 ·

Basic reporting

All comments have been taken in consideration. No further modifications are necessary.

Experimental design

All comments have been taken in consideration. No further modifications are necessary.

Validity of the findings

All comments have been taken in consideration. No further modifications are necessary.

Additional comments

All comments have been taken in consideration. No further modifications are necessary.